

# Ungulates rely less on visual cues, but more on adapting movement behaviour, when searching for forage

Jan A. Venter[1], Herbert H.T. Prins[2,3], Alla Mashanova[4] and Rob Slotow[3]

[1] School of Natural Resource Management, Nelson Mandela Metropolitan University, George, Western Cape Province, South Africa
[2] Resource Ecology Group, Wageningen University, Wageningen, The Netherlands
[3] School of Life Sciences, University of Kwazulu-Natal, Durban, South Africa
[4] Department of Biological and Environmental Sciences, University of Hertfordshire, Hatfield, United Kingdom

## ABSTRACT

Finding suitable forage patches in a heterogeneous landscape, where patches change dynamically both spatially and temporally could be challenging to large herbivores, especially if they have no *a priori* knowledge of the location of the patches. We tested whether three large grazing herbivores with a variety of different traits improve their efficiency when foraging at a heterogeneous habitat patch scale by using visual cues to gain *a priori* knowledge about potential higher value foraging patches. For each species (zebra (*Equus burchelli*), red hartebeest (*Alcelaphus buselaphus* subspecies *camaa*) and eland (*Tragelaphus oryx*)), we used step lengths and directionality of movement to infer whether they were using visual cues to find suitable forage patches at a habitat patch scale. Step lengths were significantly longer for all species when moving to non-visible patches than to visible patches, but all movements showed little directionality. Of the three species, zebra movements were the most directional. Red hartebeest had the shortest step lengths and zebra the longest. We conclude that these large grazing herbivores may not exclusively use visual cues when foraging at a habitat patch scale, but would rather adapt their movement behaviour, mainly step length, to the heterogeneity of the specific landscape.

## INTRODUCTION

African ecosystems are well known for their exceptional diversity of large mammalian herbivores, of which a large proportion are ruminant bovids with a few non-ruminant equids (*Grange et al., 2004*). The feeding type, body size and mouth morphology of large herbivores are intrinsic constraints on the habitat that they can effectively use, and provide an understanding as to how one species may be more or less constrained than another in a particular set of environmental conditions. Different species of large herbivores may use a range of different behaviours to enhance their foraging efficiency (*Bailey et al., 1996*; *Beekman & Prins, 1989*).

Corresponding author
Jan A. Venter,
Jan.Venter@nmmu.ac.za

Finding a forage patch in a heterogeneous landscape where patches differ in suitability poses a challenge, especially if individuals have no *a priori* knowledge of the location of the most suitable patches (*Bailey et al., 1996*; *Prins, 1996*; *Senft et al., 1987*). Large herbivores may gain *a priori* knowledge using memory (from a previous visit to the patch) (*Brooks & Harris, 2008*; *Dumont & Petit, 1998*; *Edwards et al., 1996*; *Fortin, 2003*) or through visual cues (*Edwards et al., 1997*; *Howery et al., 2000*; *Renken et al., 2008*). If the forage resource is complex (e.g., when forage patches are not well defined), or the distribution of the forage patches are likely to change continuously (e.g., when a patch is grazed or the grass sward becomes unpalatable due to ageing), then recalling the location of forage patches may be of limited value (*Edwards et al., 1997*). In such situations, heterogeneous in both space and time, the ability to recognise and assess different forage patches at a distance through visual cues, would promote foraging success (*Edwards et al., 1997*). An alternative behaviour to the use of visual cues would be adaptive search/movement behaviour (*Benhamou, 2007*; *Benhamou & Collet, 2015*; *Martin et al., 2015*). In heterogeneous environments, adaptive movement, at different scales of step lengths and directionality, e.g., a small-scale area-restricted search (within patches) mixed with a set of large more directional movements (between patches), can be a better search approach than an approach of using visual cues, especially when the forage resource is complex and in constant fluctuation.

A number of studies of forage patch location or re-visitation in large herbivores have linked movement patterns to the use of memory (*Brooks & Harris, 2008*; *Ramos-Fernandez et al., 2003*) or visual cues at finer scales (e.g., bite, feeding station, and food-patch scales) (*Howery et al., 2000*; *Laca, 1998*). However, it is not clear whether large herbivores use visual cues to find forage patches at a broader habitat patch scale. Our definition of habitat patch scale, adapted from *Owen-Smith, Fryxell & Merrill (2010)* and *Bailey et al. (1996)*, refers to a daily range at a 10-h temporal scale while feeding, walking, drinking and resting, with movement within and between habitats. We tested whether three grazing herbivore species use visual cues when foraging at the habitat patch scale. The selection of species represented differences in intrinsic traits (differences in body size, feeding type, digestive system and muzzle width) which presumably would influence their interaction with forage resources, e.g., search behaviour.

Red hartebeest (*Alcelaphus buselaphus subspecies camaa*) are considered to be predominantly selective grazers that will make use of browse under limited resource conditions (*Murray, 1993*). They are medium-sized (150 kg, average of both sexes) ruminants with a preference for grass. In Mkambati Nature Reserve, South Africa (our study area, hereafter referred to as Mkambati) they use 87% $C_4$ grasses (*Venter & Kalule-Sabiti, 2016*). In areas with much moribund vegetation, grazing ruminants such as the red hartebeest face particular constraints because nearly all vegetation biomass has a low quality, which reduces food intake rates (*Drescher et al., 2006a*; *Drescher et al., 2006b*; *Van Langevelde et al., 2008*). The hartebeest is an example of a concentrate selector; its muzzle width and length is specially adapted (long and narrow) to be very selective at times when good forage is scarce (*Schuette et al., 1998*). Eland (*Tragelaphus oryx*) are considered to be mixed feeders preferring browse (*Hofmann & Stewart, 1972*; *Watson & Owen-Smith, 2000*) and in Mkambati their diet consists of 79% $C_3$ forage

(*Venter & Kalule-Sabiti, 2016*). They are ruminants with a large body size (511 kg, average of both sexes) (*Venter et al., 2014b*). Zebra (*Equus burchelli*) are non-ruminants and they are much more tolerant to poor quality forage but must maintain a high rate of intake to be able to survive on this type of food (*Bell, 1971*; *Okello, Wishitemi & Muhoro, 2002*; *Van Soest, 1982*). They are mainly grazers with their diet consisting of 89% $C_4$ grasses in Mkambati (*Venter & Kalule-Sabiti, 2016*). They are medium sized (235 kg, average male and female) equids (*Venter et al., 2014b*) with a wide muzzle classifying them as bulk grazers (*Bell, 1971*).

We developed and tested predictions based on directionality (an indication if a number of turning angles, i.e., the absolute angle between movement $i$ and movement $i+1$, from a series of movements are uniform, i.e., highly concentrated in one direction, or not), step length (distance between two consecutive fixes from GPS telemetry data), and success (outcome of the search movement, whether animals arrived in better forage or not) under three patch visibility classes. In particular, we expected more directional movements with longer step lengths when animals moved to visible patches and less directional movements with shorter step lengths to non-visible patches. We expected, if animals used visual cues, that there would be longer step lengths which are more directional when they move to better forage, because they could anticipate success. No difference in step length or directionality when comparing the outcome of movements (successful and not successful) would indicate that visual cues are not used at this particular scale because then the animal did not adapt the movement (walking straight towards a observed patch) to anticipated success to find better forage. Due to the different intrinsic constraints that different species of ungulates have to deal with, we expected that each species would approach its forage search strategy in different ways regardless of the use of visual cues or not or because one species could be using visual cues more than another. Demonstrating a difference in movement behaviour between visible and not visible habitat patches, and successful or not successful movements would enable an understanding of the importance of visual cues to different large herbivore species when moving between patches at a habitat patch scale.

## METHODS

### Study area

Mkambati is a 77 km$^2$ provincial nature reserve situated on the east coast of the Eastern Cape Province, South Africa (31°13′–31°20′S and 29°55′–30°04′E). The climate is mildly sub-tropical with a relatively high humidity (*De Villiers & Costello, 2013*). The coastal location, adjacent to the warm Agulhas Current, causes minimal variation in mean daily temperatures (18 °C winter and 22 °C summer) (*De Villiers & Costello, 2013*). The average rainfall is 1,200 mm, with most precipitation in spring and summer (September–February) (*Shackleton, 1990*). The high rainfall, mild temperatures, and presence of abundant streams and wetlands provide a landscape that is not water-limited in any season. More than 80% of Mkambati consists of Pondoland–Natal Sandstone Coastal Sourveld grassland (*Mucina et al., 2006*). Forests occur in small patches (mostly in fire refuge areas) (*Mucina et al., 2006*). Mkambati contains a range of large herbivore species, but no large predators (*Venter et al., 2014b*).

The grassland is considered to be nutrient poor (*Shackleton et al., 1991*; *Shackleton & Mentis, 1992*). Grassland fire stimulates temporary regrowth high in crude protein (8.6% compared to 4.6%, in older grassland), phosphorus concentrations (0.1% compared to 0.05%, in older grassland) and dry matter digestibility (38.6% compared to 27.1%, in older grassland) (*Shackleton, 1989*). Nutrient concentrations remain elevated for up to 6 months post-burn, after which they are comparable to surrounding, unburnt grassland (*Shackleton & Mentis, 1992*). Frequent fires cause a landscape mosaic of nutrient-rich burnt patches within a matrix of older, moribund grassland. This landscape is thus continuously changing due to new fires that are set and the maturing process of the grassland. Recalling the location of grazing forage patches (using memory) would in this case be of limited value which enabled us to test predictions of movement behaviour relative to visibility of forage patches.

## Data collection

Five plains zebra (four female and one male), six red hartebeest (five females and one male) and five eland (three females and two males) were fitted with GPS-UHF collars (Africa Wildlife Tracking CC.; Pretoria, RSA) between September 2008 and July 2012. These species represented a range of intrinsic constraints which could potentially influence their foraging strategies and subsequent search movement behaviour (*Venter & Kalule-Sabiti, 2016*; *Venter et al., 2014a*; *Venter et al., 2015*). All animals were darted by an experienced wildlife veterinarian from a Robinson 44 helicopter. The work was approved by, and conducted in strict accordance with the recommendations in the approved standard protocols of the Animal Ethics Sub-committee of the University of KwaZulu-Natal (Approval number 012/09/Animal). All field work was conducted by, or under the supervision of, the first author while he was a staff member of the Eastern Cape Parks and Tourism Agency, as part of the operational activities of the appointed management authority of Mkambati (Eastern Cape Parks and Tourism Agency Act no. 2 of 2010, Eastern Cape Province, South Africa). The zebra and red hartebeest were in separate harems or herds when they were collared, but some eland (two females) were in the same herd. The collars were set to take a GPS reading every 30 min, and data were downloaded via UHF radio signal. The collars remained functional between 4 and 16 months depending on various factors, including loss of animals to poaching, natural mortality, or malfunctioning. Data downloaded from the collars were converted to geographical information system (GIS) format and sections of the data sets with missing values were removed and not used in the analysis.

Step lengths (the distance travelled between each 30 min GPS fix) were calculated for each "walk" using the Hawths Analysis Tools extension (*Beyer, 2007*) to ArcGIS (ArcGIS Desktop: release 10; Redlands, CA: Environmental Systems Research Institute). Walks were extracted per species (Eland $n = 312$; Red hartebeest $n = 309$; Plains zebra $n = 279$). A walk consisted of 20 consecutive steps lengths which constituted 10 h of movement behaviour during daylight hours (6:00 AM–6:00 PM) (Fig. 1). Ten hours of movement represented movement between patches at a habitat patch scale. To confirm whether ten hours of movement were indeed within a realistic distance range for the habitat patch scale in our situation, we compared the mean distance between patches to the mean animal walk distances

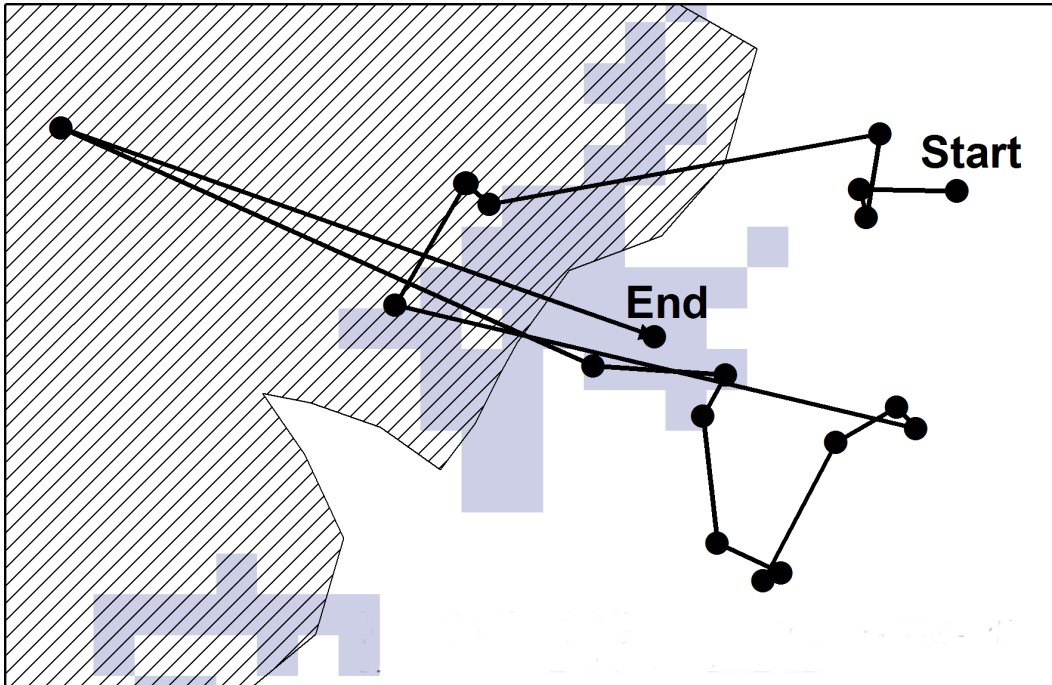

**Figure 1 A hypothetical example of a "walk" extracted for the study.** Walks were extracted from the data which included the departure point (indicated by "Start") to where the animal ended (indicated by "End"). Here the animal hypothetically spent the majority of the last three hours of its "walk" in an area which was not visible from the starting point (indicated by grey). The striped area indicates a recent fire patch.

per species. Starting points for each walk were randomly selected (by day), with the visibility from the starting point of each walk being determined using the "viewshed analysis tool" in the Spatial Analyst extension of ArcGIS (ArcGIS Desktop: release 10; Redlands, CA: Environmental Systems Research Institute). This resulted in a grid map (raster) layer that indicated all areas that were visible and not visible to the animal from that specific starting point at its shoulder height (female shoulder height: eland $\bar{x} = 1,500$ mm (*Posselt, 1963*); red hartebeest $\bar{x} = 1,250$ mm (*Stuart & Stuart, 2007*); plains zebra $\bar{x} = 1,338$ mm (*Skinner & Chimimba, 2005*)) (Fig. 1). The end point was defined by the patch where the animal spent the majority ($\geq 50\%$) of the final 3 h (7 locations) of the "walk" (Fig. 1). All patches in the landscape were allocated a unique number, and classified as either burnt grassland (fire patches) or unburnt grassland (unburnt patches) (Fig. 1). The location of the fire patches were recorded by field rangers between January 2007 and July 2012, and later digitally defined on maps using ArcGIS. Each GPS locality along a "walk" was linked to a patch classification using the Spatial Analyst extension of ArcGIS (ArcGIS Desktop: release 10; Redlands, CA: Environmental Systems Research Institute). All unburnt areas (areas that were never noted as burnt between January 2007 and July 2012) were considered as one unburnt patch, and was given the same unique identification number. The "walks" were then classified into three different visibility classes which could be a movement: (a)

to within the same patch where the starting point was located; (b) to a new patch that was visible from the starting point; and (c) to a new patch not visible from the starting point.

When an animal, at the end of a walk, ended up in: (a) a better forage patch, we considered the movement as successful; (b) the same quality patch, we considered it as no change; and (c) a worse patch, we considered it as unsuccessful. Forage quality was better in recently burnt (<6 months post fire) grassland, see *Shackleton & Mentis (1992)*, compared to older grassland. All step lengths <6 m were excluded during analysis in order to remove non-movements, as well as false movements due to GPS-error.

### Data analysis

We tested whether there was excessive variability amongst individual animal walk distances, which could potentially influence the models, by comparing mean walk distance for different species to inter-patch distances using visual inspection of box plots. This was done using IBM SPSS Statistics for Windows, Version 23.0. (IBM Corp. Released 2014, Armonk, NY).

We used the Rayleigh test of circular uniformity from CircSTats package in R (*R Development Core Team, 2011*) to calculate the mean resultant length $r$ for each individual "walk". This parameter $r$ provided a measure of the concentration of turning angles ranging between 0 and 1 (*Duffy et al., 2011*). When $r$ is close to 1, data are highly concentrated in one direction, and when it is close to 0, data are widely dispersed (*Duffy et al., 2011*). The Rayleigh test provides $p$-values associated with $r$ to test whether it is reasonable to reject angle uniformity. When $r \geq 0.5$ and the $p$-value indicated significance ($p < 0.05$), walks were considered to be concentrated in one direction (directional).

We used a linear mixed model (LMM) to assess the effect of a number of factors on mean step length per "walk". The fixed effects were species, visibility class and search outcome (success). The random effect was the individual animal. A Wald test was used to determine whether variation in step length between individuals was significant and should be included as a random effect. We did not include interactions between the fixed effects as they were not significant when included in the model. Therefore, we used post-hoc pairwise comparisons with a Bonferroni correction to determine differences in the main effects (IBM SPSS Statistics for Windows, Version 23.0, IBM Corp. Released 2014).

## RESULTS

Median walk distances for red hartebeest 2,120 m (1,305–3,068 m), eland 3,328 m (2,374–4,341 m) and zebra 3,771 m (2,255–6,755 m) were similar to distances between patches 4,994 m (2,978–7,371 m) (values in brackets give the first and the third quartiles), indicating that walks represented movements at a landscape scale as defined by (*Bailey et al. (1996)* and *Owen-Smith, Fryxell & Merrill (2010)* (Fig. 2).

A low proportion of walks in each visibility class were directional for red hartebeest (6% to not visible; 3% to visible; 8% within visible) and eland (7% to not visible; 0% to visible; 5% within visible, Fig. 3). Zebra had a higher proportion of directional walks (12% to not visible; 17% to visible; 17% within visible) than eland and red hartebeest (Fig. 3).

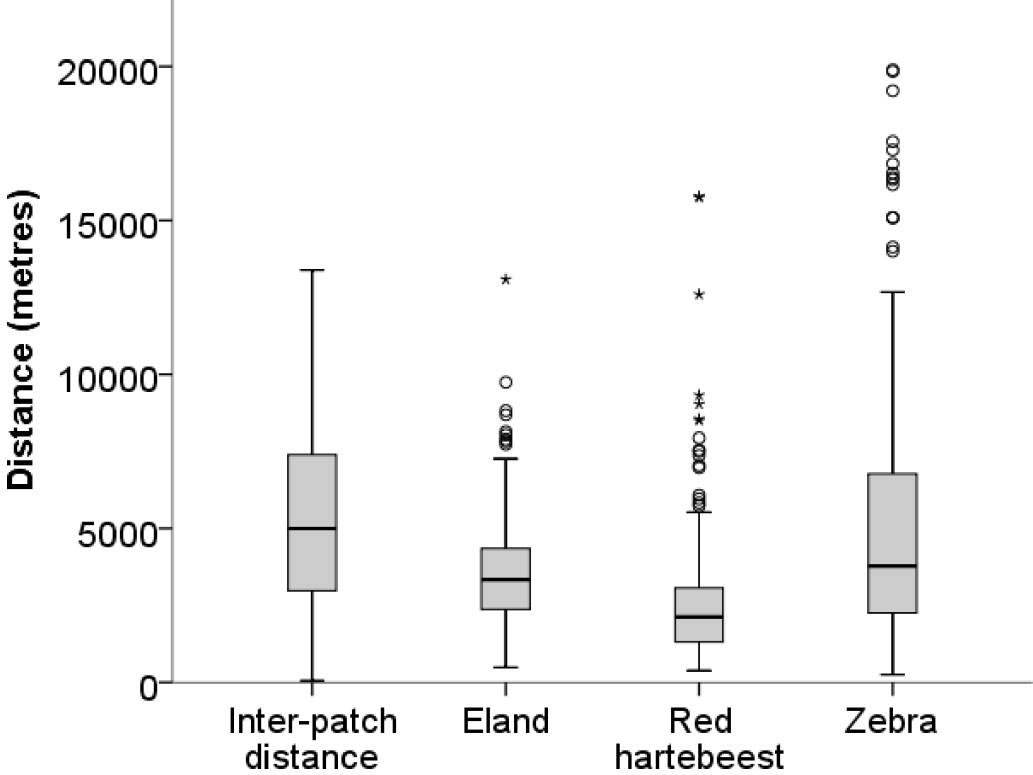

**Figure 2** **Inter-patch distances and distances moved in a 10-h walk by red hartebeest, eland and zebra in Mkambati Nature Reserve.** The horizontal line indicates the median, boxes show the first and third quartiles, vertical lines indicate 1.5 × IQR (interquartile range), circles show outliers more than 1.5 × IQR, and asterisks show outliers more than 3 × IQR.

A linear mixed model with step lengths as the dependent variable, success, visibility class and species as fixed effects, and animal ID as a random effect suggests that all fixed effects are significant (p-values 0.045, <0.0005 and 0.005, respectively). The Wald test suggests that there is a significant variation in step length between individuals ($P = 0.026$). We therefore kept animal ID in the model as a random factor. With search movement outcome, the difference between "no change" versus both "successful" and "not successful" were marginally non-significant ($p = 0.054$ and $p = 0.074$, respectively) (Table 1, Fig. 4A). Zebra had significantly longer step lengths than red hartebeest ($p = 0.005$) and nearly significantly longer step lengths than eland ($p = 0.06$) (Table 1, Fig. 4B). The difference between eland and hartebeest was not significant ($p = 0.69$) (Table 1, Fig. 4B). For visibility classes, step lengths in the "within visible" and "to visible" classes were not different ($p = 0.37$), but the step lengths for both these categories were significantly shorter than step lengths to "not visible" classes ($p = 0.002$ and $p < 0.0005$, respectively) (Table 1, Fig. 4B).

## DISCUSSION

In our study we observed little directional movement when animals (from all species) moved to visible patches, which supports a view that large herbivores do not rely exclusively on

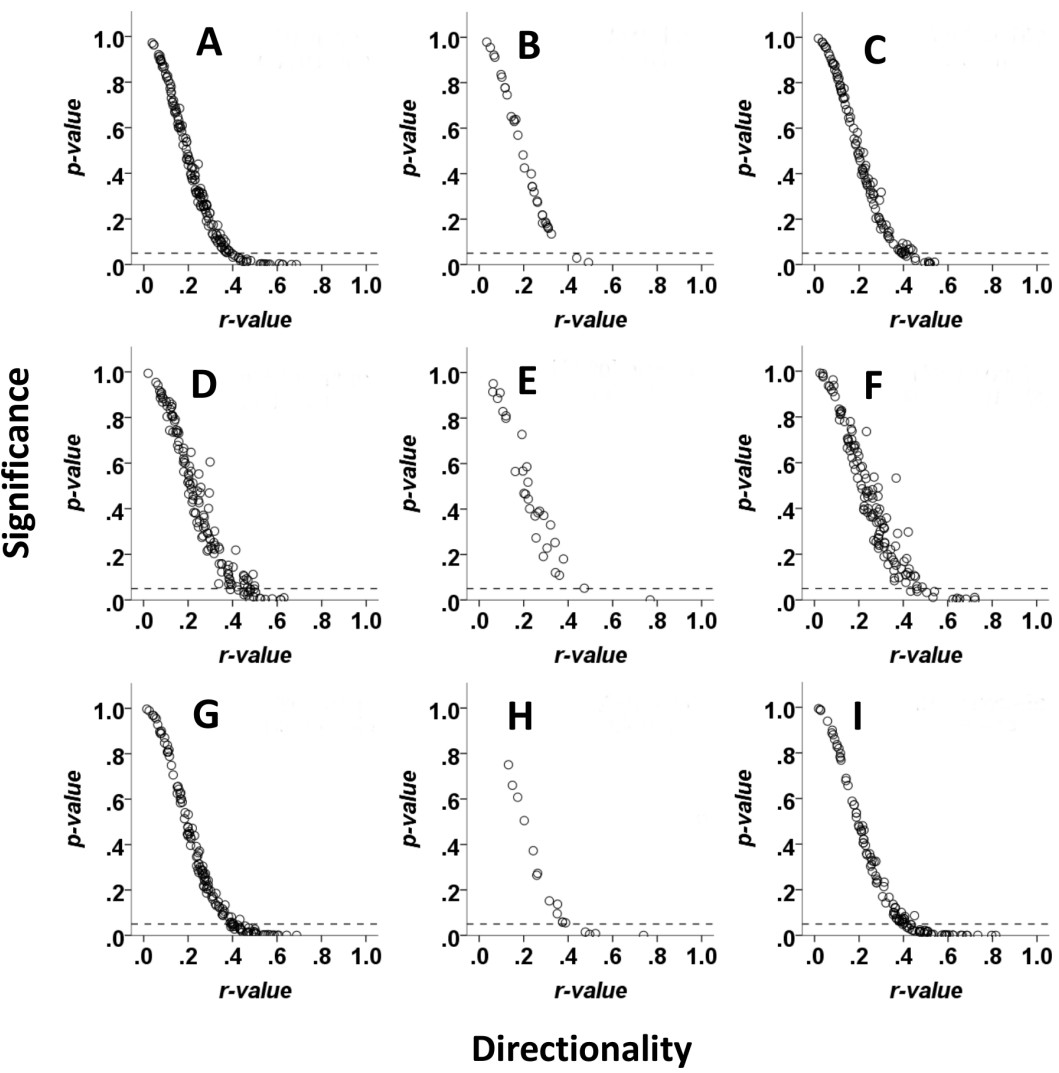

**Figure 3** **Directionality of movement of red hartebeest, eland and zebra in relation to visibility of the final location in Mkambati Nature Reserve.** Each point represents the *r* and associated *p*-value from a Rayleigh test for a single 10-h walk to locations in different patches that were not visible from the start (left column), to locations in the same patch that were visible from the start (middle column), and to locations in different patches that were not visible from the start (right column). Data are shown for eland (A–C), hartebeest (D–F) and zebra (G–I).

visual cues when moving to search for patches at a habitat patch scale. Our results support the simulations by *Benhamou (2007)* which showed that, in patchy environments adaptive movements combining small-scale area-restricted searches (within good forage patches) and large directional movements between patches (in our case, movement to forage patches which were not visible) were used as an optimal strategy to search for habitat patches. However we did not observe a clear pattern in the directionality of the movements (more directional movements between patches) which could indicate that our study animal's strategy could potentially not be as optimal a search strategy compared to the *Benhamou (2007)* simulations.
**Table 1** The results of the pairwise comparisons between species, visibility movement class and search movement outcome.

| Factor | Mean difference | Std. error | df | Sig. |
|---|---|---|---|---|
| *Species* | | | | |
| Eland vs red hartebeest | 30.505 | 24.531 | 17.737 | 0.69 |
| Eland vs zebra | −64.331 | 25.029 | 16.69 | 0.06 |
| Red hartebeest* Zebra | −94.835 | 25.068 | 16.497 | **0.005**[**] |
| *Search movement outcome* | | | | |
| Successful vs no change | 40.801 | 17.202 | 880.753 | 0.054 |
| Successful vs not successful | 2.367 | 12.681 | 874.701 | 1 |
| No change vs not successful | −38.434 | 17.094 | 879.703 | 0.074 |
| *Visibility movement class* | | | | |
| To not visible vs to visible | 89.509 | 16.214 | 873.165 | **<0.0005**[***] |
| To not visible vs within visible | 54.408 | 15.837 | 877.081 | **0.002**[**] |
| To visible vs within visible | −35.102 | 22.758 | 881.966 | 0.37 |

**Notes.**
[*]Significance: <0.05.
[**]Significance: <0.005.
[***]Significance: 0.0005.

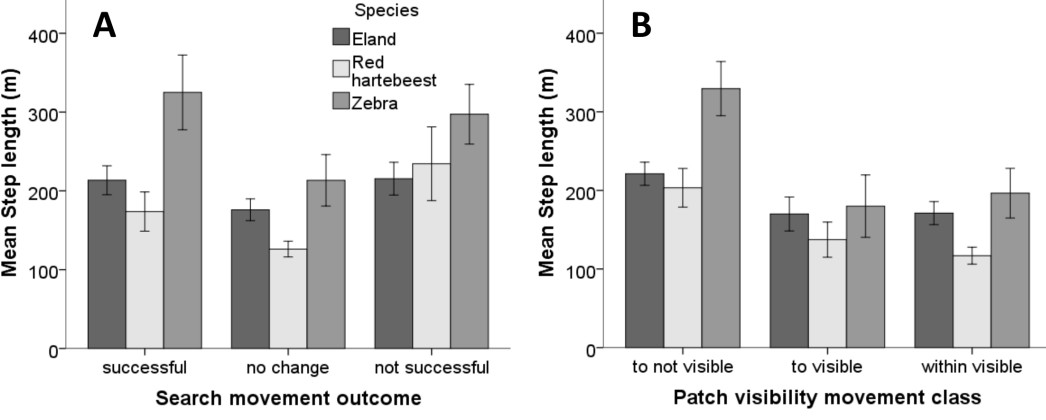

**Figure 4 Mean step length of search movement outcomes and patch visibility classes for three herbivore species in Mkambati Nature Reserve.** The relationship of (A) search movement outcome (success) and (B) patch visibility movement classes with the mean step length of zebra, red hartebeest and eland studied in Mkambati Nature Reserve. Error bars indicate 95% CI.

During fine scale search modes at the bite, feeding station and food patch scale, as defined by *Owen-Smith, Fryxell & Merrill (2010)*, animals would make use of visual and olfactory cues to find suitable forage items (*Edwards et al., 1997*; *Laca, 1998*). At coarser scales (e.g., habitat patch scale), herbivores would randomly move with larger step lengths until they are able to detect more suitable forage (at the finer scale). The search patterns displayed by our study animals thus indicate an adaption of their movement to the patchiness of the environment rather than long and directional step lengths, as expected if visual cues (or the lack thereof) had played a major role (*Benhamou, 2007*; *Benhamou & Collet, 2015*).

Adaptations of animal movement behaviour to patchiness at the habitat scale has been observed elsewhere (*De Knegt et al., 2007*; *Duffy et al., 2011*), and is supported by this study.

Red hartebeest had the shortest step lengths of the three study species. Red hartebeest is an example of a concentrate selector; its skull morphology is specially adapted to be very selective at times when good forage is scarce (*Schuette et al., 1998*). In areas with much moribund vegetation, grazing ruminants such as the red hartebeest face particular constraints because nearly all vegetation biomass has a low quality, which reduces food intake rates (*Drescher et al., 2006a*; *Drescher et al., 2006b*; *Van Langevelde et al., 2008*). By being more selective, hartebeest would probably need to have more spatially complex movement scales. Red hartebeest, being the smaller ruminant (compared to eland), needing less, but better quality forage to meet their nutritional and energy requirements (*Demment & Soest, 1985*; *Illius & Gordon, 1992*), used a strategy where they foraged using smaller and less directional steps (compared to zebra), whether they were moving within patches or to visible patches, but increased their step lengths when moving to not visible patches, just like zebra and eland. The smaller step lengths could be explained by their tendency to move slower and spend more time in less nutritious patches which was observed by *Venter et al. (2014a)*. They could thus be more effective in extracting more nutritious material from older moribund grass tufts (due to their adapted muzzle) and therefore be moving in shorter more concentrated steps. In addition, because they are ruminants, they probably spend a significant amount of time ruminating, and moving less, compared to a non-ruminant such as zebra.

Eland had shorter step lengths than zebra but slightly larger step lengths than red hartebeest. Eland is one of the larger African ruminant species and is considered to be a selective feeder (which includes browse) that require a diet of high nutritive value, low fibre and high protein content (*Arman & Hopcraft, 1975*). In Mkambati they primarily use browse and make little use of grass as forage (*Venter & Kalule-Sabiti, 2016*). They also have a relatively small rumen in relation to their body size and retain food in the rumen for a shorter time (shorter compared to cattle), which allowing for a greater consumption rate (compared to hartebeest) (*Arman & Hopcraft, 1975*). Zebra (non-ruminant) and eland (ruminant) have different body sizes but have similar digestive capacity due to differences in their digestive system (*Demment & Soest, 1985*). It is, therefore, surprising that eland has shorter step lengths than zebra. This behaviour could possibly be linked to their diet, as being able to browse they can overcome the challenge of dealing with a landscape of nutrient poor, moribund grassland by eating forbs and trees (when available). Forbs are common, especially in newly burned patches in Mkambati (*Shackleton, 1989*). Because trees are a resource that does not change as rapidly as continuously burnt grassland, eland should be able to return to browsing patches using memory. This could possibly explain their movement behaviour, although one would have expected more directional movements if memory were being used.

Zebra used larger step lengths and had more directional walks (although still a small proportion of their walks) compared to the eland and hartebeest. These variations could be linked to differences in the species' intrinsic traits, such as digestive system, muzzle width and body weight (*Prins & Van Langevelde, 2008*; *Senft et al., 1987*). Zebra, a non-ruminant, is less efficient at digesting food and has to maintain a higher intake-rate to maintain

its energy requirements (*Bell, 1971*; *Demment & Soest, 1985*; *Illius & Gordon, 1992*). This should cause them to move more frequently from one food patch to another as food patches are depleted due to grazing (*Bell, 1971*). In addition, they have a wider muzzle than the two ruminant species which makes them capable of using very short grass swards (which are common in recently burned grass patches). Zebra have been shown to prefer newly burned grassland (*Sensenig, Demment & Laca, 2010*), but they are forced to keep moving to new food patches because the lower biomass in a given patch is depleted much more quickly (*Venter et al., 2014a*). In addition, the overall higher directionality of zebra movement could indicate that they are more efficient in finding new forage patches. Both these factors would cause movements with larger step lengths and more directionality, as we observed with this species.

There is a certain degree of uncertainty whether walk directionality was affected by the step length. *Hurford (2009)* showed that GPS measurement errors might lead to reporting overly tortuous movement when the distances between locations were smaller than 20 m. Although we removed all distances smaller than 6 m from the analysis there is a chance that part of our turning angle measurements were affected by GPS error. For example, the larger proportion of short steps in hartebeest might explain why directionality in hartebeest movements was smaller than we expected.

Our study provides some evidence indicating that large grazers might not exclusively rely on visual cues when foraging at a habitat patch scale, but rather adapt their search mode, mainly longer step lengths, when they move to not visible patches. The animals used this adaptive approach to foraging to cope with continuously changing forage conditions. In addition, it shows that different species search for forage in different ways, which could indicate that search strategies are linked to intrinsic traits such as body size, feeding type, digestive strategy and muzzle width.

## ACKNOWLEDGEMENTS

Mkambati Nature Reserve staff, students from the University of Kwazulu-Natal and students from Pennsylvania State University, Parks and People program for providing field assistance.

### Funding

This study was financially supported by the University of Kwazulu-Natal and Eastern Cape Parks and Tourism Agency. The funders had no role in study design, data collection and analysis, decision to publish, or preparation of the manuscript.

### Grant Disclosures

The following grant information was disclosed by the authors:
University of Kwazulu-Natal and Eastern Cape Parks.
Tourism Agency.

## Competing Interests

The authors declare there are no competing interests.

## Author Contributions

- Jan A. Venter conceived and designed the experiments, performed the experiments, analyzed the data, wrote the paper, prepared figures and/or tables, reviewed drafts of the paper.
- Herbert H.T. Prins and Rob Slotow conceived and designed the experiments, reviewed drafts of the paper.
- Alla Mashanova analyzed the data, contributed reagents/materials/analysis tools, reviewed drafts of the paper.

## Animal Ethics

The following information was supplied relating to ethical approvals (i.e., approving body and any reference numbers):

The work was approved by, and conducted in strict accordance with the recommendations in the approved standard protocols of the Animal Ethics Sub-committee of the University of KwaZulu-Natal (Approval number 012/09/Animal). All field work was conducted by, or under the supervision of, the first author while he was a staff member of the Eastern Cape Parks and Tourism Agency, as part of the operational activities of the appointed management authority of Mkambati (Eastern Cape Parks and Tourism Agency Act no. 2 of 2010, Eastern Cape Province, South Africa).

## Data Availability

The raw data has been supplied as Supplementary File.

## Supplemental Information

Supplemental information for this article can be found online at http://dx.doi.org/10.7717/peerj.3178#supplemental-information.

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
