# Peer review of "Ungulates rely less on visual cues, but more on adapting movement behaviour, when searching for forage"

_PeerJ, doi:10.7717/peerj.3178_

## Round 0.1 · original submission · Major Revisions

Reviewer 1 identifies a major concern that, by considering step length as the distance traveled during 30-min intervals rather than distance moved between changes in direction, the authors have failed to meet a fundamental assumption of the models being tested. This reviewer further notes that he/she had provided the same critique in a review for another journal 2 years ago. Two other fundamental potential problems were noted, the mixing of search and patch exploitation within individual 10-h "walks" and pooling data among individuals. Despite these problems, the reviewer recommends major revision rather than rejection because the data set is large and important and because it is possible that a sound publication could be produced by either returning to the raw data to extract correct step lengths or by eliminating the comparison of search models from the objectives of the study.

Reviewer 2 does not indicate such fundamental problems but notes a number of cases of important information missing from the methods and potential concerns regarding some analyses.

Within the limits of my knowledge of this area, the point made by Reviewer 1 seems valid. It is unfortunate that, having had this critique, the authors apparently did not change their analysis or include in the methods a statement supporting the validity of their approach in relation to this criticism. My decision is, therefore, to request major revisions, taking into account the comments of both reviewers. Note that PeerJ does not include 'impact' or 'importance' as a criterion for publication, so a comparative analysis of move speed and direction in relation to visibility of patches would still be potentially acceptable, even without the strong connection to search theory.

In addition, the authors might consider whether parametric descriptions of variation (as in Fig. 2) is appropriate for data that seem to be not normally distributed. Finally, I have not engaged in a detailed examination of organization, style and grammar of this version, given that major rewriting is likely but I would do so on a later version.

Reviewer 1 ·

Basic reporting

On reading the manuscript I had a strong sense of déjà vu and subsequently realized that in the summer of 2014 I had reviewed two earlier versions of it for Ecosphere. I did not retain copies of these earlier versions but I do still have my reviews. I can see that the authors have not addressed my primary concerns. I pointed out significant errors in technique, fact, calculation and interpretation.

The manuscript tells a cohesive story in a concise, precise and easy to follow manner. The subject matter – effectively the relative merits of Levy walks and composite correlated random walks as models of complex movement patterns -- is topical and high profile, and is currently being played out in PNAS, Science and other top flight journals. The methods of analysis – maximum likelihood methods and the Akaike information criterion – are state-of-the-art and are well accepted by the research community. The presentation – the figures and accompany tables – are very clear and readily interpretable. The data set is extensive and impressive. There are, however, significant errors in technique, fact, calculation and interpretation. The hallmark of a Levy walk is a step-length distribution with a heavy power-law tail where a ‘step’ is a movement bout made between consecutive, significant turns (aka ‘re-orientation events’). The authors’ analyses appear to be based on distances travelled between consecutive locational fixes made at regular time intervals that are set by the arbitrary data sampling protocol and so without biological significance. Such analyses cannot be used to test for the presence of Levy walks. Indeed, it has long been realized that turns should be used to define steps more generally; in, for example, the parameterization of correlated random walks of the type used by the authors (see Turchin's classic book- Quantitative analysis of movement: Measuring and modelling population redistribution in animals and plants, 1998). The problem with the authors method is readily understood. Suppose for simplicity that an animal walks in a perfectly straight line but walks with variable speed. If the speed variations are exponential, then the 'step' lengths (as defined by the authors) will be exponential-like and the movement pattern will be classified as being a correlated random walk. If speed is intermittent (the animal occasionally stops) then the steps will appear to be multiphasic and the walk will be classified as being a composite correlated random walk. An analysis based on turns would, however, correctly identify the movement as being a straight-line.

The steps defined by the authors can be used to assess some differences in the movement patterns for the different visibility classes but these differences cannot, as the authors have done, be framed: in the context of Levy walks; expectations of Levy search theory and; the relative merits of Levy walks and composite correlated random walks (multiphasic walks) as models of the movement pattern data. The language adopted by the authors is highly misleading and will cause confusion. The authors' have a choice. They either re-do their analyses by determining steps in a way that allows them to properly test to Levy walks, composite walks etc. [the relatively coarse sampling may preclude this possibility]. Or they dispense with relating their results to such models and instead state their results more directly and simply, i.e., by stating that animals have longer, straighter movements when targets are not visible; and that foragers adapt to conditions encountered during searching by switching from an extensive to an intensive mode of searching upon detection of a patch…

There are potentially two additional significant problems. First, a search ends when a patch is located. This truncation will give any step-length distribution a high exponential tail. This makes the detection of intrinsic search patterns, like Levy walks difficult, unless the patches are sparsely distributed so that intrinsic scaling extends over a large range of scales. This problem is compounded by the fact that the foragers could switch from an extensive Levy walk or extensive ballistic search to an intensive Brownian search upon entering a patch, making whole movement patterns resemble composite correlated random walks with multi-exponential step-length distributions. This is just the kind of the distribution found to best fit the movement pattern data. Second, the authors have pooled data for a number of different individuals. The resulting step-length distributions (which tend to be multi-exponentials) could reflect intrinsic variability that is displayed by all individuals, or reflect variability amongst individuals, or some combination of these two possibilities. The authors need to discriminate between these possibilities.

Some of the referencing and terminology are decidedly odd and suggests that the authors are not sufficient familiar with the relevant literature. Some examples:

L56 The authors state that they classify random movement behaviours (i.e., patterns) as being random walks (which is a tautology) or as being Levy walks. What the authors actually attempted to do was to classify the observed movement patterns as being either simple random walks, multiphasic random walks, or Levy walks. Moreover, the bracketed term “Brownian motion” is problematic because all movement patterns become Brownian at significantly long scales because of truncation at patches, at the boundaries of the home range, at topographical features etc.

L59 Viswanathan et al. 1999 is an appropriate reference for Levy walks but not for finite-specific random walks. Turchin’s classic book is an obvious reference.

L60 Edwards et al. 2012 iconoclastic paper is far from being an appropriate reference for Levy walks.

L60-63. This is too glib because the relative merits of any search pattern will depend on lots of factors, including the initial conditions of the search, i.e., whether searching begins close to a patch or midway between adjacent patches. See e.g., Viswanathan et al. 1999.

L65-74. Here the authors make much of the fact that composite correlated random walks can resemble Levy walks. This is true but it calls for an explanation because most composite correlated random walks will not resemble a Levy walks. The resemblance requires fine-tuning of the parameters in the multiphasic walk and this suggests selection for Levy walk characteristics. This in turn suggests that the arguments about the relative merits of Levy walks and composite correlated random walks are somewhat misplaced.

Experimental design

The dataset is impressive

Validity of the findings

The data analysis is flawed fundamentally.

Additional comments

I am recommending major revision rather than outright rejection because the dataset is impressive and could yield interesting information if analysed in an appropriate manner.

Reviewer 2 ·

Basic reporting

From the point of view of the language, this paper is well written. The overall structure is appropriate and the raw data is provided. However, there are a few points that could be modified to improve the reporting: I would recommend introducing in the introduction the concept of adaptive movement behaviour, to change the final part of the intro to put the hypothesis before the predictions, the results and methods should be slightly changed so that they relate more directly to the questions/hypothesis examined and table 4 is not necessary, so it could be taken out (see further details in comment to author).

Experimental design

The research is original in that it compares movement across different species of ungulates within the same area o study, and tries to depict the effect of visual cues to drive their movement. The data collection and selection are carefully done. The statistical analyses are mostly appropriate, although some improvements could be performed (see further details in comment to author).

Validity of the findings

The results are valid, relate to the questions initially raised. The discussion is consistent with the results. The main weak point is the fact that the comparison among species (according to traits like diet, etc…) is not properly presented in the introduction.

Additional comments

This paper presents and interesting comparison of ungulate movement among different visibility scenarios, trying to disentangle the effect of visual cues from other movement behaviours. The detail comparison of several ungulate species in the same study area is especially valuable and the analyses are appropriate. However, I would suggest the following changes before continuing with the publication process:

• Line 22-23 from abstract, Brownian motion could/should be replaced by composite Brownian movement.
• Introduction: I think the introduction would benefit from introducing the “adaptive movement behaviour” concept as alternative or other alternative movement behaviours beyond the one relying on visual cues or memory cues.
• I missed hypothesis about the comparison among species. By reading the introduction on, doesn’t get the clear idea that later there will be so much time devoted to comparing the three species.
• Lines 84-86 seem like the hypothesis to me and I would suggest including them before the predictions (i.e. lines 82-83).
• I would recommend including the sections “study area”, “methods” (change to “data collection”) and “data analyses” all as subheadings of the heading “methods”.
• In lines 127-129: the authors mentioned that they eliminated sections with missing values. Did they test for the special autocorrelation of these values? Are the always link to certain type of vegetation? If so specify.
• Are radio-collared animals always females? Specify sex.
• Lines 156 – 159: are the “walks” classified considering the difference in patch between the start and end position? Clarify in the text.
• Line 165: if I understood well, what they are comparing here is the mean walk distance PER INDIVIDUAL for different species? If so specify.
• Lines 184-191: I am not aware of the exact algorithms used to calculate “r”, but is this method considering the intrinsic correlation between step length and turning angles in the case of GPS data. The existent of a measure error in GPS data leads by default to higher turning angles when step lengths are shorter than when they are longer. See for example (Hurford, A. (2009) GPS measurement error gives rise to spurious, turning angles and strong directional biases in animal movement data. PLoS ONE, 4, e5632.). This could explain the fact that Zebra has more directional steps (just because they are larger steps)). Did the authors take this into account?
• The paper could benefit from ANOVA’s being replaced by Linear mixed models in which the single walks could be used as variable response and “the individual ID” is introduced as random factor, to account for non-independence of the data. The “herd” could be also added as random factor to account for the case of the eland with several marked individuals moving together.
• In general results should be adapted a bit more to the more direct answering of the questions raised in the introduction.
• Line 205: do you mean figure 3?.
• Table 4 is not necessary, because all the information given here is already in table 5, so I suggest removing and highlighting in “bold” the preferred models in table 5.
• Lines 221-226, add unnecessary length to the text. I suggest removing them.
• Line 230 (do you mean figure?).
• Lines 284-287: wouldn’t that mean that animals have more directional movements in the case of moving to a non-visible patch? Unfortunately that was not the case in your results... maybe rethink this explanation.
• In legend form table, the authors mentioned table 6.3, but I didn’t find this table in the manuscript.
• Table 5: for coherence sake the results of the different visibility scenarios, could be presented in the same order as in the other tables (first within patch,etc…)
• Figure 5: replace “visable” by “visible”.

---

## Round 0.2 · Minor Revisions

Overview

Reviewer 1 is satisfied with the revised presentation but notes the inappropriate citation of a study whose conclusions are no longer considered valid. (This point might be addressed by also citing the critical subsequent article.) Reviewer 2 is also basically satisfied with this version, but has several additional recommendations. My reading of the manuscript identified a number of issues, including some that overlapped with those of Reviewer 2. These include insufficient discussion of some conclusions and numerous problems with the presentation, including awkward, wordy or unclear sentence construction, errors in grammar and punctuation, and mistakes in the references. I have listed the non-grammatical issues below and have indicated the grammatical errors and potential alternative wording using highlighting and inserted comments on the pdf.

Regarding my comments below, you may treat them as a third review, making appropriate changes if my suggestions are valid and providing a clear rebuttal if they are not.

Editor's comments

Primary concerns
1) The conclusion that ungulates adapt their movement patterns to the demands of their foraging environment appears in your title, abstract, discussion and conclusions. However, I could not find any clear development of the evidence supporting this claim. The brief assertion regarding the value of random movement is not explicitly linked to your data, nor is there a discussion of possible alternative explanations for any randomness in the observed patterns. Is it possible that this conclusion is a residue from the previous version of your manuscript and needs to be limited to a paragraph discussing this as a possible explanation?
2) The conclusion that visual cues are unimportant to the discovery of new patches is supported by the shorter step lengths and lack of directionality when moving to visible patches. However, I think you need to summarize the evidence more explicitly and address any inconsistencies in the data and possible alternative explanations for the observed patterns, i.e., a more rigorous critical discussion.
3) Most of the discussion is devoted to a reasonable discussion of species differences. However, the conclusion (L264-266) that traits such as body size and morphology, feeding type, etc. play a role is overstated. While species differences are well supported, there is no rigorous analysis to show which characteristics influence movement patterns, nor can there be with only 3 species. As a major empirical finding of your study, a summary of the species differences ought to be included in the abstract.
4) Success in finding a new patch appears in methods and results, but I did not notice a mention in the Introduction, predictions, or Discussion. What is the role of this measure?

Other concerns
Check all references in text; sometimes the parentheses should be around only the year rather than the full reference (e.g., L124, 125, 178).
It is reasonable to put quotation marks around "walk" at first use. After that, they are not needed, and your use throughout the text and figure captions is inconsistent.

L20. Define step length, even in the Abstract, because it may not be familiar to all readers and the expression is sometimes used for stride length, potentially creating confusion.
L51. Incomplete comparison: more optimal than what?
L63. This is a good place to define what you mean by both directionality and step length to avoid any confusion.
L64ff. Table 1 would be more appropriate at the start of the Discussion, rather than putting the results of your hypothesis test before you have even provided the methods. Also, this section needs a clearer justification of the predictions and an inclusion of the within-patch movement predictions.
L102. Rather than just refer to the existence of constraints, wouldn't it be a good idea to provide some details of what they are for each species and how they might affect foraging? Perhaps these details could be presented in the Introduction following L83. Indicate which herbivore species were selected for study and how they varied in ways important to foraging. The details of sexes of collared animals would remain in Methods.
L109-153. I found this critical section somewhat confusing. I think that the order of presentation could be improved for greater clarity. My proposal below is a suggestion that should indicate how I was confused, but still may not be the most logical order.
• Start by stating that the distance traveled between 30-min readings was considered a step and 20 consecutive steps during daylight hours was considered a walk.
• Then, provide an explicit statement defining step length and direction and walk length (is it the sum of steps or distance between first and last locations).
• You don't mention direction in this section, yet refer to it early in the analysis section; is walk direction also used?
• Follow this with how you tested that 20 steps provided a suitable scale. This is unclear because you defined habitat patch scale in the Introduction (L60) and now introduce landscape scale. It is not clear whether you are using two terms for the same scale (as a reader might infer from the repetition of references) or whether you are trying to clarify whether they are in fact the same.
You refer to the last 3 h as 6 final locations (L135) whereas I would have expected 7 locations for 6 steps or 3 h.
I don't understand the random starting point issue because random starting points are not possible for 20 consecutive steps within daylight.
You need to join the information on visibility estimation with the information on their application to final patches given in the next paragraph. The sentence describing this classification (L48-51) is awkward and needs to be rewritten. Were final patches classified as visible or not from the starting point?
The concept of successful and unsuccessful walks could be clearer. Did everything hinge on the 6-month burn criterion, so all patches were defined as good (<6 mo.) or poor (> 6 mo.) and walks were defined as successful if they switched from poor to good and unsuccessful if they switched from good to poor? Couldn't a search be successful if switched from good to good (for example, if the grass had been heavily cropped in the original patch?) Otherwise, an animal in a good patch could not make a successful move.
L148-151. This sentence is not very clear and the order of classes does not match the way it was presented in the results. I suggest something like "The walks were then classified into different visibility classes: a) movements in which the end point was within the same patch as the starting location, b) movements in which the end point was a different patch visible from the starting location, and c) movements in which the end point was a different patch not visible from the departure point. Figure 3 should match this order. Note that I tried to use terms consistent with your previous definition of end point. Also, if movements within the same patch always have visible end points as implied by Fig. 3, you should state that here.
L175. Confusing. Not sure what an 'error bar plot' is or what you are trying to say. Do you mean to say something like 'Mean walk distances (detailed values) were similar to distances between patches (value), indicating that walks represented movements at a landscape scale'? Note also the confusion between habitat patch scale in the Introduction and landscape scale here and in the Methods noted above. Also, I don't think that you defined inter-patch distance in your methods. Is it the distance between patch centers or nearest edges? Finally, as I noted in my comments on the previous version, if you are going to use parametric (mean, SD) measures for your comparison, you should confirm that the data are normally distributed. Otherwise, use nonparametric measures.
L188. The Wald test was not mentioned in data analysis section of Methods.
L190. Again, order of factors discussed should correspond in methods and results text, as well as table and figures for ease of understanding by readers. Several of your p-values approach significance, and I think this should be mentioned because it suggests a possible effect. The statement on the effect of visibility class is unclear. You did not clearly indicate that although the factor was statistically significant, though marginally, in the overall analysis, none of the two-way comparisons were statistically significant. Again, it may be important that two did approach significance, especially as there was an overall treatment effect.
L214. As noted above, you need to provide a justification for your assertion that adaptation of movement to patchiness at a habitat patch scale is confirmed by your study. First, confirmation or 'proof' is a bit strong for normal philosophy of science. Second, a correlational study showing that relatively few 20-step sequences are directional seems to provide limited evidence for such a complex assertion. Given that a walk is expected to have a mix of directional and non-directional movements, what proportion of significant r-values would it require to reject this hypothesis?
L229. You are referring to movement intensity and complexity, but I don't think you have defined these terms or demonstrated how they apply to zebras.
L240-241. The logic of using less and more nutritious patches in a similar way based on a comparison with zebra for within patch and to non-visible patch movements is unclear. This needs elaboration. The verb 'relates to' is too vague. How are these observations related?
L258. Did you explain what you mean by less complex movement and how that applies to eland?
L261ff. What is evidence for adaptation of foraging mode to heterogeneity and quality? There is definitely no evidence for the effect of morphological traits and diet (with only 3 species differing in multiple traits, insufficient evidence to support role of any one or combination of traits).
References need to be checked carefully. There are capitals that should be lower case in some article titles and the opposite in some book titles, some missing italics for scientific names, and a book title in which the publisher information is unclear.

Fig. 1. I think that the figure would be strengthened by more detail. If it is a real as opposed to a hypothetical case, you could specify the species and date. A distance scale bar on the figure would also help. I only count 18 location points rather than the 21 I expected for 20 steps. If my expectation is incorrect, something in the methods may need clarification.

Fig. 2 needs some work. You don't need to interpret the results in the caption because that was done in the text. The caption is imprecise because it implies that the figure only shows the mean. I suggest "Mean +/- 1 S.D. (m) of inter-patch distances and distances moved in a 10-h walk by three species of grazing herbivore in Mkambati Nature Reserve)." You don't need the second x-axis label 'inter-patch distance and species'. You do need a hyphen for 'inter-patch' and to remove 'mean' from the y-axis label.

Fig. 3. The caption is incomplete and somewhat confusing. I suggest the following. Also, note that for greater readability, you should have consistency in the order of end point types between the methods and caption/results.
"Directionality of movement of three species of herbivore in relation to visibility of the final location in Mkambati Nature Reserve. Each point represents the r and associated p-value from a Rayleigh test for a single 10-h walk to locations in different patches that were not visible from the start (left column), to locations in the same patch that were visible from the start (middle column), and to locations in different patches that were not visible from the start (right column). Data are shown for eland (A,B,C), hartebeest (D,E,F) and zebra (G,H,I)."

Fig. 4. I proposed some changes to refer to the relationship rather than effect because this is a correlational study. Note that relating the figure to the text will be easier if panels A and B and the order of movement outcomes and visibility classes correspond to the order used in the results, Table 2, (and preferably the definitions in the methods as well).

Reviewer 1 ·

Basic reporting

No comment

Experimental design

No comment

Validity of the findings

No comment

Additional comments

The revised manuscript is considerably better than the original submission, as it is free of misunderstandings about Levy walks, misunderstandings which are being perpetuated in the literature. In their rebuttal the authors disagree with me about the use of “step-lengths” when defined as being the distances between (arbitrarily defined) positional fixes, and cite several studies which have used this inappropriate specification. I will not labour the point but suggest that the authors (1) read Viswanathan et al. (1999) which they cite, and (2) take a look at the much-cited comprehensive review of Levy walks by Zaburaev et al. (Rev. Mod. Phys. 87, 483, 2015). The relevant subsection is “VI. Levy walks in biology. iii Levy walks vs Levy flights”. These leading experts conclude rightly that the misuse of step-lengths as defined above is partially responsible for causing the controversies surrounding Levy walks and does not add positively to the on-going debate about the biological relevance of Levy walks. I hope that the authors will take note of this in the future.

The citing of Viswanathan et al. (1999) is not really appropriate because the data analysis in that paper was overturned by Edwards et al. (2007).

Reviewer 2 ·

Basic reporting

The paper is well written and the focus and structure ar much clearer and consistent now.

Experimental design

The data collection and experimental collection are correct and the statistical analyses are much more appropiate now.

Validity of the findings

The results are sound and robust althought the discussion would benefit from highlighting the fact that the results sometimes did not support the hypothesis and then link to the explainations of "why" already included...

Additional comments

Comments to the Author


I thank the authors for addressing the comments made by the reviewers, including mine, and for making the changes they did. Overall, I feel the manuscript has improved importantly by making these changes. Nevertheless, I still have a few more specific comments:

1) The introduction is now much better and appropriate. However I am still missing some words about the predictions regarding the differences among species given that so much weight is given later to the differences among species. I understand that the inclusion of the first paragraph was meant to fulfil my request. However, this does not replace a specific prediction about the differences among species… the prediction does not have to be specific to the species involved necessarily, but if could deal with the expectations of different movements type according to species types.
2) I am not entirely convinced about the response about the autocorrelation of missing values. I still think the autocorrelation of missing values or the relation of missing values to a certain habitat should be checked and mentioned at least in an appendix. I assume that if the missing values are biased per habitat or species, this could affect the results. If that there is no bias please justify/clarify in more detail.
3) Regarding the turning angles “problem”. I think that even though the authors corrected partially for it by eliminating the step lengths below 6 meters. I still would acknowledge this “potential bias” in the discussion to partially explain the potential influence of this “error” in the results.
4) I personally do not understand the need to do the “Pairwise comparisons was done using a Bonferoni”. Using the fitted values from the LMM (figure 4). All combination of comparison can be done directly by comparing the the 95%CI of one category overlaps with the mean of the other category or not as a post-hoc test. Is this what the authors meant? This would be a more elegant option because it take into account the differences while controlling by the other variables included in the model.
5) The Wald test is first mentioned in the results but not in the methods. It should be included in the methods too or eliminated from the results.

---

## Round 0.3 · accepted · Accept

I now consider the manuscript suitable for publication following a few minor corrections which can be fixed in productio .

L19 remove parenthesis before zebra
L61 comma after 'However'
L69,78,81 Provide scientific names at first use here and removed them from the later paragraph (L135ff)
L92 replace 'landed up' with 'arrived'
L101 insert 'that' after constraints
L104,268,317 replace 'invisible' with 'not visible'
L164 I would prefer to see a statement describing how distance between patches was calculated after this sentence (for example, whether it measures distance between centers or nearest edges), but it is not essential if the author prefers not to do so.
L184 comma after patch
L185 insert 'quality patch' and comma after 'same'
L185 insert 'a' before worse, comma after patch
L206 insert 'A' before Wald
L214 delete extra 3771 m
L215 insert space between ')('
L216 insert space before 'and'
L219, 221 remove 'and' after semicolon (3 places)
L220 replace ')(' with comma and space
L222 dependent (spelling)
L224 remove spaces around hyphen in p-value
L227 replace 'marginally significant' with 'marginally non-significant'
L229 replace 'approached significance for' with 'and 'nearly significantly longer step lengths than'
L233 comma after 0.0005
L256 replace 'confirmed' by 'supported', as agreed on item 21 of the rebuttal letter
L339 lower case foraging hypothesis
Fig. 2 We agreed by email that the caption would be changed to: “Inter-patch distances and distances moved in a 10-h walk by red hartebeest, eland and zebra in Mkambati Nature Reserve. The horizontal line indicates the median, boxes show the first and third quartiles, vertical lines indicate 1.5 x IQR (interquartile range), circles show outliers more than 1.5 x IQR, and asterisks show outliers more than 3 x IQR.”